# Endophytic Fungi Residing within *Cornus florida* L. in Mid-Tennessee: Phylogenetic Diversity, Enzymatic Properties, and Potential Role in Plant Health

**DOI:** 10.3390/plants13091250

**Published:** 2024-04-30

**Authors:** Asha Maheshwari, Margaret T. Mmbaga

**Affiliations:** 1Department of Agricultural and Environmental Sciences, Tennessee State University, Nashville, TN 37209, USA; ashamaheshwari163@gmail.com; 2Pharmacia, Nashville, TN 37209, USA

**Keywords:** biocontrol, endophytes, flowering dogwood, ornamental plants, plant pathogens, plant protection

## Abstract

Endophytic fungi that reside internally in healthy, asymptomatic plants often benefit their hosts by promoting plant growth and/or providing plant protection against abiotic and biotic stresses. However, only a small fraction of the estimated 1.5 million fungal endophytes have been identified. In this study, a total of 369 isolates of fungal endophytes in 59 distinct taxa were isolated from stem samples of *Cornus florida* (flowering dogwood). All isolates belonged to species of phyla Ascomycota and Basidiomycota distributed across five orders and 11 genera. Isolates belonging to the same family clustered together in a phylogenetic tree generated from a cluster analysis using MEGA 7 software. Diversity indices of the fungi revealed a rich and diverse community that included several species associated with leaf spots, blight, cankers, and/or dieback diseases. Pathogenicity tests confirmed 16 fungal endophytes as *C. florida* pathogens, including some well-known destructive pathogens *Botryosphaera dothidea*, *Colletotrichum acutatum*, and *C. gleosporoides*. Isolates of the fungal endophytes possess the capacity to produce extracellular hydrolytic enzymes (cellulase, amylase, pectinase, laccase, chitinase, and protease) that are known to function in tissue penetration, plant colonization, nutrient acquisition, and disease suppression in both plant pathogens and endophytes These results support the interchangeable pathogenic–endophytic roles for some taxa.

## 1. Introduction

Endophytic fungi are ubiquitous and comprise a highly diverse polyphyletic group of microorganisms that reside internally in plant tissues, which are often asymptomatic for at least part of their life cycle [1,2,3,4]. Endophytes exist in diverse ecological relationships within their host, ranging from symbiosis to antagonism against host pathogens. Currently, endophytes have been examined in approximately 300,000 plant species, and almost all the plants were found to harbor at least one or more endophytes [5,6,7]. It is estimated that 1.5 million fungal species reside in plants, but only 70,000–100,000 fungal species have been identified, which represents only 7% of the fungal endophytes [8,9]. These previous studies indicate the need for more intensive studies to identify the fungal endophytes associated with plant species under diverse environments and geographic locations. Fungal endophytes colonizing medicinal plants have been more commonly studied [5,6,10,11,12,13,14]. Endophytes associated with ornamental plants remain largely unexplored, and this study examined an ornamental tree commonly known as flowering dogwood (*Cornus florida* L.). that is native to the northeastern and southeastern region of the United States. Although *C. florida* is a well-known ornamental plant with high economic importance [15,16], the study focused on a limited area in Mid-Tennessee where planting is in large nurseries, forest undergrowth, landscaping, and backyard gardens. Traditionally, *C. florida* has been used in treatment of malaria [16,17] and *Cornus* spp. are known to possess antimicrobial, anticancer, and antidiabetic properties [16,17,18,19,20]. Furthermore, numerous pharmacological benefits, phytochemicals, and therapeutic applications have been associated with this *Cornus* [16,17]. In contrast, the microbial diversity, and their biological potential of *Cornus* endophytes remains largely unexplored. The association of pathogenic fungi in foliar diseases of *Cornus*, such as dogwood anthracnose caused by *Discula destructive* Redlin [21,22] and powdery mildew caused by *Erysiphe pulchra* [15,23] have been well studied. Although reports are available on beneficial epiphytic fungi associated with *C. stolonifera* (red-osier dogwood) in British Columbia [24], *C. controversa* (giant dogwood) [25] and *C. florida* in middle Tennessee [26], information is lacking on the diversity of endophytic microorganism that reside in *C. florida* and their potential role in plant protection.

Biodiversity studies have demonstrated that some endophytes are saprophytes, while others are latent or opportunistic pathogens that may become quite aggressive when the host plant is stressed and the environment is favorable to disease development [27,28]. It has been hypothesized that some fungal endophytes, which grow without causing disease in their host plants, may have evolved from pathogenic strains [29,30,31]. Numerous studies have shown that endophytic fungi can play an important role in protecting plants against pathogens, promoting plant growth, and increasing host resistance to abiotic stress [4,31,32,33]. Thus, for plant health management, it is critical to understand the microbial diversity of endophytes and their potential as pathogens or biocontrol agents. Endophytic fungi have been shown to produce various metabolites including hydrolytic enzymes, such as cellulases, amylases, chitinase, protease, xylanases, etc. [4,34,35,36,37]. These enzymes are known to play a role in tissue penetration/colonization, nutrient acquisition, and disease suppression that benefit both plant pathogens and biological control agents. The objectives of this study were to (a) understand the diversity of fungal endophytes that reside in *C. florida*, (b) evaluate the potential role of these fungal endophytes as plant pathogens and biological control agents against selected diseases, and (c) explore the potential of these fungal endophytes to produce extracellular enzymes that may benefit plant health.

## 2. Results

### 2.1. Isolation of Endophytic Fungi

A total of 1080 tissue segments of stem samples were processed from 72 healthy and asymptomatic *C. florida* trees collected at eight different locations in middle Tennessee (Table 1). A total of 369 fungi were identified using ITS sequences. The highest frequency of fungal endophytes was recorded in samples collected from the Murfreesboro location (H, 52%) while the lowest isolation frequency was observed at a McMinnville location (D, 21.3%). Analyses of ITS sequences using BLASTn showed that 96.5% of all fungal endophytes recovered from this study belonged to the phyla Ascomycota and 0.03% belonged to Basidiomycota, while 3.47% of the fungi could not be identified. These fungal endophytes were distributed over four classes, 12 orders, 21 families, 37 genera, and 39 species. In this study, the relative abundance of all fungal endophytes, along with class and family, are summarized in Appendix A. Fungal endophytes in the Xylariales were most frequently isolated and overall, the fungi at different locations were diverse (Figure 1). Some fungi were dominant with significant differences in the relative abundance within each taxonomic group, as was observed in Sordariomycetes (Figure 2).

### 2.2. Phylogenetic Analysis of Dogwood Fungal Endophytes

PCR amplification of rDNA ITS region generated DNA amplicons that ranged from 450 to 650 bp in size and resulted in a total of 217 ITS sequences. Based on ITS sequence data, the stem samples collected from eight locations yielded 59 distinct fungal taxa. Out of these, 39 taxa from 254 isolates were recognized to the species level, 13 taxa from 84 isolates were identified to the genus level, 4 taxa (26 isolates) to the order level, 1 taxon (2 isolates) to the class level, and two unidentified taxa resulted from 3 isolates. Phylogenetic analysis provided evidence of evolutionary relationships among the endophytic fungi associated with *C. florida*. The evolutionary history inferred using the neighbor-joining method according to Saitou and Nei, (1987) [38] is presented in Figure 3. The optimal tree with the sum of branch length = 5.82767659 is shown in Figure 3. The percentage of replicate trees in which the associated taxa clustered together in the bootstrap test (1000 replicates) are shown next to the branches (Felsenstein, 1985) [39]. The tree was drawn to scale, with branch lengths in the same units as those of the evolutionary distances used to infer the phylogenetic tree. The evolutionary distances computed using the p-distance method (Nei and Kumar, 2000) [40] are in the units of the number of base differences per site. The analysis involved 216 polymorphic nucleotide sequences. All the ambiguous positions were removed for each sequence pair. A total of 1853 positions were available in the final dataset. The tree was rooted with the ITS sequence of Caloscypha fulgens (DQ491483) of class Pezizomycetes. The bootstrap values of >70% are shown on the branches.

All identifiable taxa of fungal endophytes isolated from *C. florida* were contained in two clades representing Ascomycota and Basidiomycota in the phylogenetic tree. Within the Ascomycota clade, endophytic fungal isolates were classified into five orders within the class Sordariomycetes (Xylariales, Diaporthales, Glomerellales, Trichosphaeriales, and Xylomelasma), four orders within the class Dothideomycetes (Dothideales, Botryosphaeriales, Pleosporales, and Capnodiales), and one order within the class Leotiomycetes (Phacidiales). These results indicated that the endophytic fungi from flowering dogwood were highly diverse, as shown by the relative abundance of fungi in different orders and classes within each order (Figure 1 and Figure 2). A smaller group of endophytic fungi belonging to the Basidiomycota was represented by two orders within the class Agaricomycetes (Polyporales and Russulales).

Out of 163 endophytic fungi, 117 were identified at a 99% sequence similarity threshold, which consisted of 44 distinct genotypes (Appendix A). Two endophytic fungi could not be categorized to any genus or species as no matches were found in the GenBank database, leading to the assumption these fungi may represent species for which ITS sequences have not been deposited in the GenBank database or novel, undescribed fungal species (unidentified spp.). Overall, Xylariales (43.9%) and Pleosporales (34.9%) were the most abundant and diverse groups at all locations (Appendix A and Figure 1). *Didymosphaeria variabile* was reported as the most dominant species, with a relative frequency (RF) of 11.1% followed by *Hypoxylon perforatum* (RF: 5.4%) *Pestalotiopsis microspore* (RF: 5.2%), and *Daldinia childiae* (RF: 4.6%) (Appendix A). The composition of endophytic fungal species in *C. florida* communities showed that 35 fungal species were exclusively isolated from one or two locations and the abundance of dominant species seems to differ among different locations (Appendix A).

### 2.3. Diversity Indices for Endophytic Fungi in C. florida

The values obtained from the Shannon Weiner diversity index (H’), Simpson’s diversity index (1-D), and Margalef’s richness (Dmg) illustrated biodiversity of endophytic fungi in stem samples of *C. florida* at specified locations. Pielou’s evenness indices (J) predicting differences in endophytic fungal populations at different locations ranged between 0.85 to 0.95 and J values closer to 1 represent evenness of endophytic fungal species within the communities (Table 2). The highest J value was in the site H (Murfreesboro, Rutherford County), where the number of tissue segments sampled was lowest (Table 1). The species richness at different locations, as calculated using Menhinick’s index presented in Table 2, had its highest value at site E (Dmn = 3.046) and lowest at site D (Dmn = 2.12), both in Warren County, McMinville. Camargo’s index expressed site-specific fungal dominance of 0.75 (highest) for Site B samples (McMinnville, Warren County, TN, USA) and ranged between 0.328–0.62 for other locations. (Table 2). These tests indicate that endophytic fungi in *C. florida* stems are quite diverse and differ among different locations. Further, Sorenson’s similarity index (QC) revealed the highest similarity (QC = 0.304) was between Davidson County (Nashville, TN, USA) sites F and G, and the lowest similarity (QC = 0.053) was between Warren County (McMinnville, TN, USA) site E and Davidson County (Nashville, TN, USA) site G, which also represented the farthest distance between locations (Table 2 and Table 3).

### 2.4. Pathogenicity Testing

When the endophytic fungal strains were tested for their ability to produce disease symptoms in *C. florida*, the majority did not show any symptoms indicating that they were nonpathogenic on leaves. However, 16 isolates caused clearly defined necrosis indicating that these endophytes, which were derived from symptomless plants, could behave as latent pathogens and foliar disease could develop when the environment was suited to disease development (Table 4). Inoculations with fungal endophytes, such as *Alternaria alternata, Nemania serpens*, *Pestalotiopsis microspora*, *and Botryosphaeria dothidea*, resulted in severe symptoms of necrotic lesions that rapidly spread across the entire leaf surfaces within 7–10 days (Figure 4). Similar results were obtained in all the replications, and the same inoculated fungi were re-isolated, and the results were confirmed by repetition of the experiment. A list of endophytic fungi that were nonpathogenic and did not display disease reactions on *Cornus florida* leaves during pathogenicity tests is presented in Table 5.

### 2.5. Screening for Extracellular Enzymes

Based on the results of pathogenicity tests, a total of 50 endophytic fungi were screened for the production of hydrolytic enzymes, such as cellulase, amylase, laccase, pectinase, chitinase, and protease (Figure 5a–f). Of these, 49 endophytic fungi displayed the capacity to produce two or more enzymes under study. The results showed that four endophytic fungi, *Nemania* sp. (BA13F1), *Rosellinia corticium* (B-A17F2), *N. serpens* (D-A34F2), and *D. variabile* (FA62F1), were able to produce all six enzymes that were assayed (Appendix A). Based on the enzymatic index (EI), which was calculated as discussed in materials and methods, endophytes were categorized as weak, moderate, or strong producers of secreted enzymes, or non-producers of secreted enzymes in the test assays. Out of 50 endophytic fungi evaluated, more than 75% produced cellulases as scored by hydrolysis zone around the colony after staining with congo red (Figure 5a). Of these, 27 endophytes were weak producers of secreted cellulase enzyme, four were moderate producers, and seven were strong producers of secreted cellulases (Appendix A). The highest cellulase activity was observed with *Seimatosporium lichenicola* (E-A47F6, EI: 5.4) and the weakest cellulase activity was observed with *N. serpens* (D-A34F2, EI: 0.02). Secreted amylase enzyme was produced by 55% of endophytic fungi (Figure 5b) with EI values ranging from 4.0 to 0.02 (Appendix A). Pectinase and chitinase activity were observed in 48% and 52% of the endophytic fungi tested, respectively (Appendix A). Higher activity for pectinase (Figure 5c) was observed in *Pestaliopsis mangifera* (G-A68F2) as compared to other fungal endophytes. Of the endophytic fungi screened, 96% produced a secreted protease at varying levels (Figure 5d). Out of 50 isolates evaluated for the production of secreted protease, 37 fungi were weak, six were moderate, and five were strong producers of secreted protease. The maximum EI for secreted protease was 3.7 by *R. corticium* (B-A17F2), followed by 2.8 for *Nemania* sp. (B-A13F1); and 2.5 for *N. serpens* (D-A34F2) (Appendix A). None of the endophytic fungi was a moderate or strong producer of extracellular chitinase (Figure 5e). The maximum chitinase activity with EI of 0.6 was shown by *Hypoxylon* sp. (A-A4F2) followed by 0.2 from *R. corticium* (B-A17F2) (Appendix A). The production of secreted laccase was indicated by a color change from a colorless culture medium to a blue color due to the oxidation of 1-napthol (Figure 5f). Secreted laccase activity was exhibited by 65% of the endophytic fungi evaluated. The highest secreted laccase activity was exhibited by E-A47F6 (*S*. *lichenicola*) with EI of 3.5, followed by A-A7F1 (*Seimatosporium lichenicola*) with 3.0.

## 3. Discussion

*Cornus florida,* is one of the most economically important ornamental plants, with great ethnobotanical value [15,17]. In this study, diverse endophytic fungi were isolated from stems of trees that did not display disease symptoms, and the diversity of endophytic fungi within *C*. *florida* was revealed for the first time. The fungal genera recovered in this study that have also been previously reported as endophytes in other plant species include *Hypoxylon*, *Pestalotiopsis*, *Xylaria*, *Cytospora*, *Diaporthe*, *Daldinia*, *Colletotrichum*, *Nigrospora*, *Botryosphaeria*, *Diplodia*, *Alternaria*, *Phoma*, *Cladosporium*, and *Peniophora* [12,13,14,27,41,42,43]. Because previous studies by Huang et al. [10] reported that fungal endophytes are more frequent in stems as compared to other plant tissues, this study focused on fungal endophytes in the stem of flowering dogwood, and unveiled a rich and diverse community of fungal endophytes associated with *C. florida* stems. More studies are needed to advance our understanding of the role of diverse endophytes in plant health.

ITS sequence-based analysis has previously been reported as a reliable method for fungal identification in numerous studies. The ITS-based phylogenetic analysis presented herein represents the first such study for fungal endophytes in *C. florida*. Based on the ITS sequence data, the stem samples collected from eight locations yielded 59 distinct taxa of fungal endophytes. Out of these, 39 taxa (254 isolates) were recognized to the species level, 13 taxa (84 isolates) to the genus level, 4 taxa (26 isolates) to the order level, 1 taxon (2 isolates) to the class level, and two were unidentified taxa (3 isolates). These results confirmed the diverse range of the endophytic fungal species associated with *C. florida*. Most of the endophytic fungi belonged to phylum Ascomycota, and this finding is consistent with other studies on diverse plant species [12,13,42,44]. The low prevalence of the phylum Basidiomycota in *C. florida* is also consistent with previous reports on other plant species [14,42,45]. Sordariomycetes and Dothideomycetes were the most dominant and diverse classes of endophytes found in this study, which is similar to endophytic fungal communities recovered from *Glycine max* [41] and *Ocimum sanctum* [12]. Futhermore, fungi within the Xylariales and Pleosporales orders were major endophytic colonizers of *C. florida* and comprised more than 75% of the total isolates.

In this study, the number of tissue segments used in the isolation of fungal endophytes was similar (150) among sites A–F, but lower in sites G (107) and H (75). However, the relative abundance of isolates recovered varied among sampling sites and was highest at sampling site F followed by Site E, and lowest at site A (Table 1). The total number of tissues sampled can be used as a measure of sampling effort, and more species are likely to be encountered with more intensive sampling. However, the total number of species detected varied among the different locations, ranging between 1.34 and 3.05 as determined by Menhinick’s index (Dmn) [12,46]. It was observed that a higher isolation frequency does not necessarily result in higher diversity in any sampling site. Species diversity indices at different locations (Table 2) were evaluated using the Shannon–Wiener index (H´), Simpson’s diversity index (1-D), and Margalef’s index (Dmg) [11,12,44,46,47]. Species diversity reflects how many different types of taxa are present in communities, and considers both species richness as well as the dominance/evenness of the species. Species diversity from the Shannon–Wiener index showed moderate variation among sampling sites (1.89–2.67), while analysis of Margalef’s index showed high variation among different locations. Simpson’s Index is a measure of dominance and weights toward the abundance of the most common taxa, and Simpson’s diversity index was uniform. The fungal dominance assessed by Camargo’s index (1/Dmn), where Dmn is species richness [12,47] and species richness is a measure of the number of species (or other taxonomic level) present at a site (Table 2). The Pielou evenness index ranges between 0 and 1.0, with lower values reflecting more variation in abundances among different taxa within the community. Our results show the Pielou evenness index of 0.85–0.95 (Table 2).

The simplest measure of species richness is just the number of species that have more than one individual recorded per site. Thus, species richness was determined by the total number of different species recorded in a sample, at the sampled site. The species richness among different locations was determined by Menhinick’s index (Dmn) [12,46]. Overall, differences were observed in fungal endophyte frequency in *C. florida* colonization, species diversity, richness indices, and similarity indices of endophytic fungal communities residing in stems of *C. florida* at different locations. These results provide evidence that the host–endophyte interaction is greatly influenced by the micro-environment, even when two communities are in proximity. This finding is consistent with previous reports by Silva-Hughes et al. [14] and Yokoya et al. [48]; however, the fungal endophyte community could also be influenced by the host genetics and development. In addition, some species were common among all sampling sites, as reflected by Sorenson similarity indices (Table 3), thereby suggesting host affinities.

The association of fungal endophytes with latent and opportunistic fungal pathogens has been extensively studied [27,28,49,50,51]. These latent/opportunistic fungal pathogens have the ability to asymptomatically colonize plant vascular tissue and remain quiescent while the environment is not conducive to disease development. When a host plant becomes stressed and/or the environment becomes favorable for the development or pathogenicity, the quiescent fungi can become pathogenic and cause irreversible damage [5,27]. Some of the fungal endophytes isolated from *C. florida* in this study have been previously reported as agronomically important pathogens or saprophytes, such as species of the genera *Hypoxylon*, *Diaporthe*, *Diplodia*, *Cytospora*, *Colletotrichum*, *Pestalotiopsis*, *Botryosphaeria*, *Phyllosticta*, *Epicoccum*, and *Phoma*, which have been associated with leaf spots, blight, cankers, and/or dieback diseases [27,28,52,53,54,55,56,57,58,59]. Of these potential pathogens, *C. gloeosporioides* [60], *C. acutatum* [52], and *Botryosphaeria dothidea* [61] have been reported as destructive pathogens of *C. florida* and/or other *Cornus* sp.

Our results with detached leaf assays provided significant information on potential pathogens residing as endophytes or latent pathogens in healthy *C. florida*. The occurrence of these pathogens in asymptomatic plants confirms the previous reports on latent phase of several endophytes before the manifestation of disease symptoms [49,50,51,59]. Some of the endophytes confirmed to have pathogenic potential on detached leaves have not been reported as pathogens of *C. florida*, and the potential pathogenicity of these endophytes warrant further studies. In contrast, some fungal endophytes regarded as pathogens were unable to cause disease on detached leaves, hence supporting the interchangeable pathogen–endophytic existence of these fungi [7,30]. Information from this study may facilitate early response to future disease outbreaks that may adversely impact dogwood production. Isolates of fungi that are known to be pathogenic but did not display disease symptoms on pathogenicity tests indicated a possibility that they were nonpathogenic forms that existed as endophytes with potential antagonism against pathogenic forms. This phenomenon has been reported on other fungi, such as *Fusarium oxysporum*, that have both pathogenic and nonpathogenic forms and the nonpathogenic forms have been developed as biological control agents against the pathogenic forms. Further pathogenicity testing using whole plants is needed to confirm and identify potential pathogens of dogwoods that exist endophytically as latent pathogens.

Some fungal endophytes isolated in this study, such as *Nigrospora spaherica*, *Hypoxylon* sp., *Entonema* sp., etc., have previously been shown to have biological control potential against *Fusarium solani*, *F. oxysporum*, *Macrophomina phaseolina*, *Cercospora nicotianae, Phytophthora capsici*, *P. irrigata*, *P. cryptogea,* and/or *P. nicotianae* [62,63]. Furthermore, endophytic *N. sphaerica* was shown to reduce the disease severity of Phytopthora blight in pepper under greenhouse conditions [63]. Based on these results, we hypothesize that latent pathogens living endophytically within the stems of *C. florida* may be suppressed by antimicrobial metabolites produced by other endophytes living in the same ecological niches.

Out of the fungal endophytes associated with *C. florida,* 50 were examined for the ability to produce hydrolytic enzymes, such as cellulase, amylase, laccase, pectinase, chitinase, and protease. The results suggested that these fungal endophytes have the ability to catabolize a wide range of polymeric substrates (Appendix A). Understanding the pattern of substrate utilization and the ability to produce various extracellular enzymes may help to understand the diverse ecological roles of endophytic fungi [64]. Production of extracellular cellulase in 75% of the isolates and protease in 96% of the isolates were prominent, as compared to other hydrolytic enzymes, such as pectinase produced by about 48% of endophytic fungi. The role of pectinase in fungal pathogenesis, tissue penetration, and plant decomposition has been well established [65]. In this study, all fungal endophytes that caused disease symptoms on *C. florida* leaves were found to produce protease, and some of these isolates also produced pectinase activity. According to Choi et al. [35], fungal endophytes may produce pectinase and/or protease if they are latent pathogens or weak parasites. Only 55% of endophytic fungal isolates tested were able to degrade starch in this study, which contrasts with the previous report by Choi et al. [35], where all of the endophytic fungi produced amylase. In this study, over 65% of tested fungal endophytes were able to produce extracellular laccase, which can mediate the oxidation of phenolic substrates, such as lignin. These results contrast to other studies [35,36,37], which found that few or no fungal endophytes produced laccase. Many nonpathogenic fungi were able to produce chitinase enzymes, which suggests their potential in the biological control of fungal pathogens or insect pests. Extracellular enzymes secreted by endophytic fungi may have a possible role in conferring resistance to pathogen attack, tissue colonization, and/or deriving nutrition from plant as a latent pathogen [34,35,37]. Thus, the results obtained from enzyme assays further support a potential role of some endophytic fungi isolated from *C. florida* in plant protection against fungal pathogens.

## 4. Materials and Methods

### 4.1. Sample Collection

Stem samples, 5- to 10-inches (12.7 to 25.5 cm) long, were randomly selected from healthy and mature *C. florida* trees from eight distinct locations in middle Tennessee. Most samples were collected from McMinnville, Warren County because of its prominence in dogwood production within Tennessee (Table 6). The plant material was collected in sterile bags, stored at 4 °C, and processed within 24 h of sampling.

### 4.2. Isolation of Endophytic Fungi and Maintenance

Endophytic fungi were isolated using a modification of methods previously described by Schulz et al. [7]. The stem samples were cleaned thoroughly under running tap water, surface sterilized in 70% ethanol for 1m and 4% sodium hypochlorite for 3–5 m. The samples were subsequently rinsed three times with sterile distilled water for 5 m each. A heat-sterilized scalpel was used to cut the vascular tissue into 3- to 4-mm segments, 15 segments per sample were transferred into Petri dishes containing acidified potato dextrose agar (PDA; Sigma-Aldrich, St. Louis, MO, USA), incubated at room temperature, and observed for appearance of mycelial growth. Records on each isolation and its plant origin were maintained for calculation of endophyte isolation frequency. At least two or three successive sub-culturings of hyphal tips on PDA were used to obtain pure fungal cultures, and each isolate was maintained in long-term storage at three different conditions: 15% glycerol at −80 °C, 15% glycerol at 4 °C, and sterile distilled water at 4 °C. For short-term use, fungal isolates were maintained on PDA at 28 °C.

### 4.3. Molecular Identification of Fungal Endophytes

Based on the colony morphology on PDA plates, fungal isolates were grouped together and all morphotypes were selected for identification. A total of 163 representative isolates grown on PDA at 28 °C for 7–10 days were identified using DNA sequence-based methods. Total genomic DNA was extracted from ca. 200 mg of mycelia for each fungal isolate using the FastDNA kit (MP Biomedicals, Santa Ana, CA, USA) following manufacturers’ instructions. The DNA concentration and purity were determined using a NanoDrop spectrophotometer (NanodropLite, Thermo Fisher Scientific, Wilmington, DE, USA) at 260 and 280 nm. The DNA was amplified following PCR protocols and a universal primers pair, internal transcribed spacer (ITS)-1 (5′-TCCGTAGGTGAACCTGCGG-3′) and ITS-4 (5′-TCCTCCGCTTATTGATATGC-3′) [66]. Each PCR reaction was conducted in a 50-μL final volume containing 20 ng of genomic DNA, 5X PCR buffer (Promega, Madison, WI, USA), 2.5 mM of MgCl2, 250 μM dNTP, 0.4 μM of each primer, and 1.0 units of Taq DNA polymerase (Promega, Madison, WI, USA). The following PCR amplification conditions were used on Bio-Rad C1000 Touch^TM^ thermal cycler (Bio-Rad, Hercules, CA, USA): initial denaturation at 94 °C for 3 min, 30 cycles of denaturation at 94 °C for one minute, primer annealing at 57 °C for one min, extension at 72 °C for 2 min, and a final extension at 72 °C for 4 min. Each PCR product was analyzed on gel electrophoresis using 1.4% agarose gel. The amplicon was purified using Exosap (USB Affimetrix, Santa Clara, CA, USA), and submitted to Eurofin Genomics (Louisville, KY, USA) for DNA sequencing.

The taxonomic identification of fungal isolate was performed using the ITS sequences analyzed against the available sequences in GenBank National Centre of Biotechnology information (NCBI) database (https://www.ncbi.nlm.nih.gov). The identity of each organism was based on the closest similarity match in the GenBank database. Closest similarity match of ≥97% was used for species identity; sequence similarities between 95–97% were used for genus, and <95% were used for fungal order, family, phylum, or labeled as ‘unassigned’ [11,42]. To confirm the identification, phylogenetic analysis of endophytic fungi was carried out using ITS rDNA and reference sequences of closely related taxa from the GenBank. The fungus *Caloscypha fulgens* (DQ491483) of class Pezizomycetes was used as an outgroup for phylogenetic analysis. Multiple sequence alignment was performed using ClustalW version 1.6 prior to tree construction.

### 4.4. Phylogenetic Analysis of the Fungal Endophytes

ITS sequences retrieved from GenBank were used to perform phylogenetic analysis using software MEGA version 7.0 [67]. The phylogenetic tree was constructed using software MEGA version 7.0 [67]. The evolutionary history was inferred using the neighbor-joining method [38] and evolutionary distance was calculated using the p-distance method [40]. The robustness of the internal branches was assessed with 1000 bootstrap replications [39].

### 4.5. Pathogenicity Test

Based on the information on the identity of the fungal endophytes, isolates that have previously been reported as pathogens of different plant species including *Cornus* spp. were evaluated for pathogenicity on *C. florida* using a detached leaf technique. Leaves were collected from 2- to 3-year-old, healthy plants, surface disinfested using 70% alcohol for 30 s, 4% sodium hypochlorite for 3 min, and rinsed three times in sterile, distilled water. After the leaves were blotted dry using sterile paper towels to remove any moisture, they were arranged in a moist chamber (sterilized, clear-plastic box with two layers of sterile, moist paper towels). The leaves were inoculated with individual fungal endophytes using 5-mm mycelial plugs from the colony edge of 7- to 10-day old cultures growing on PDA. The inoculum was placed on the adaxial side of half of the leaf and sterile PDA plugs were placed on the symmetrical half as the control (mock inoculum) treatment. A total of 67 fungal endophytes were tested using three replicates of individual leaves per isolate and arranged in a randomized complete block experimental design. The inoculated leaves were incubated in 100% relative humidity at room temperature, and the development of disease lesions was monitored. Inoculated leaves that displayed defined necrosis were compared to control leaves at 2 weeks post-inoculation, a time when the disease expressions were clearly defined. To complete Koch’s postulates, a small piece of leaf tissue was excised from the margin of the necrotic lesions to re-isolate the pathogen on PDA. The isolation of the same endophytic fungi confirmed its pathogenicity, and the pathogenicity tests were repeated once.

### 4.6. Screening of the Endophytic Fungi for Extracellular Enzymes

Fifty different isolates of endophytic fungi, including some isolates that were shown to be pathogenic in the inoculation tests, were screened for their ability to produce extracellular enzymes using standard agar plate assays. Mycelial plugs (5-mm diameter) of 7-day-old cultures were placed on Glucose Yeast Peptone medium (1.0% peptone, 0.5% yeast extract, 2.0% D-glucose, and 1.5% agar, pH 7.2) containing specific substrates for the respective enzymes. The following substrates were used to test for the respective extracellular enzymes; carboxy methyl cellulose for cellulases, starch for amylase, pectin for pectinases, gelatin for proteases 1-napthol for laccases, and colloidal chitin for chitinase following recommendations by Sunitha et al. [37] and Choi et al. [35]. To test for enzyme activity, the Petri plates were incubated at 27 ± 2 °C for 5–10 days and flooded with 0.1% congo red to test for cellulases, Gram’s iodine for amylase, hexadecyltrimethylammonium bromide for pectinases, and acidified mercuric chloride for proteases (gelatin hydrolysis). The enzymatic index (EI) was determined for each isolate according to Choi et al. [35] in which
$$Enzymatic\;index\;\left(EI\right)\;is:\;\frac{Average\;diameter\;of\;clear\;zone-Average\;colony\;diameter}{Average\;colony\;diameter}$$

Based on the EI values, endophytic fungi were categorized into four groups according to Choi et al. [35]): non-producers (EI = 0), weak producers (EI ≥ 0, < 1), moderate producers (EI ≥ 1, < 2, and strong producers (EI ≥ 2). The experiment was performed in four replicates and repeated to confirm the results.

### 4.7. Data Analysis

The frequency of stem colonization by endophytic fungus was calculated as the number of endophytic fungal isolates recovered divided by the total number of segments plated. The relative frequency (%) was determined as tissues colonized by individual taxon divided by total number of endophytic fungi in the sample according to Raja et al. [13] and Yao et al. [43]. The species richness among different locations was determined by Menhinick’s index (Dmn) [12,46], in which Dmn = S/√N where S is the total number of different species recorded in a sample, and N is the total number of endophytic fungi in the sample. The fungal dominance was assessed by Camargo’s index (1/Dmn), where Dmn is species richness [12,47]. Species diversity in *C. florida* at different locations was evaluated using the Shannon–Wiener index (H’), Simpson’s diversity index (1 − D), and Margalef’s index (Dmg) [11,12,44,46,47] using the following equations:$$H^{\prime }=-\sum_{i}^{S}pi\;ln\left(pi\right)\;1-D=1-\sum_{i}\frac{ni\left(ni-1\right)}{N\left(N-1\right)}\;Dmg=\frac{\left(S-1\right)}{ln(N)}$$
wherein, p is the relative abundance (n/N) of individuals of one endophytic fungal species found (n) divided by the total number (N) of endophytic fungi in a sample. S is the total number of different species recorded in sample. The value for Simpson’s diversity index ranged between and 1, in which 0 means no diversity and 1 means infinite diversity. Species evenness was estimated by Plieou’s evenness index (J) estimated as follows: J = H’/ln S where H’ represents the Shannon–Wiener index of endophytic fungi in a sample and S is the total number of species observed in the sample [12], which ranges from 0 to 1, in which 0 mean no evenness and 1 means complete evenness.

Similarity among endophytic fungal communities at different locations was determined using Sorensen’s similarity coefficient (QS). QS = 2a/(2a + b + c) where ‘a’ is the number of shared species in both communities, while ‘b’ and ‘c’ are the numbers of fungal species recovered in two communities under investigation QC range from 0 (no species overlap between communities) to 1 (total species overlap between communities) [12,43].

## 5. Conclusions

This study is the first to examine the diverse endophytic fungi associated with *C. florida*. The study further noted that dogwood pathogens, such as *B. dothidea*, *C. acutatum*, and *C. gleosporoides*, can reside in *C. florida* tissues as endophytes without causing symptoms, but it is somewhat expected that disease outbreaks caused by these fungi may occur whenever host and environmental conditions become conducive to disease development. This study showed that *C. florida* fungal endophytes have the potential to secrete hydrolytic enzymes (i.e., cellulase, amylase, pectinase, laccase, chitinase and protease) that play a role in tissue penetration/colonization, nutrient acquisition, and disease suppression.

## Figures and Tables

**Figure 1 plants-13-01250-f001:**
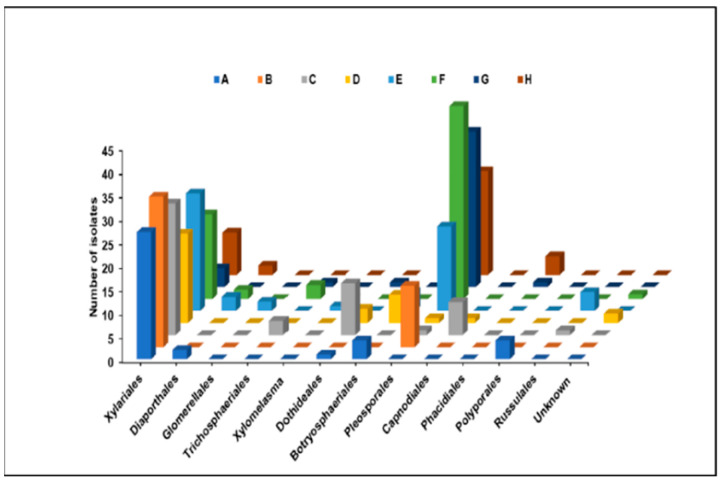
Relative abundance of endophytic fungi associated with *Cornus florida* with respect to the affiliation on the order level at different sampling locations A–H, in which location A–E are in McMinnville, F and G in Nashville, and H is a Murfreesboro location in Tennessee, USA.

**Figure 2 plants-13-01250-f002:**
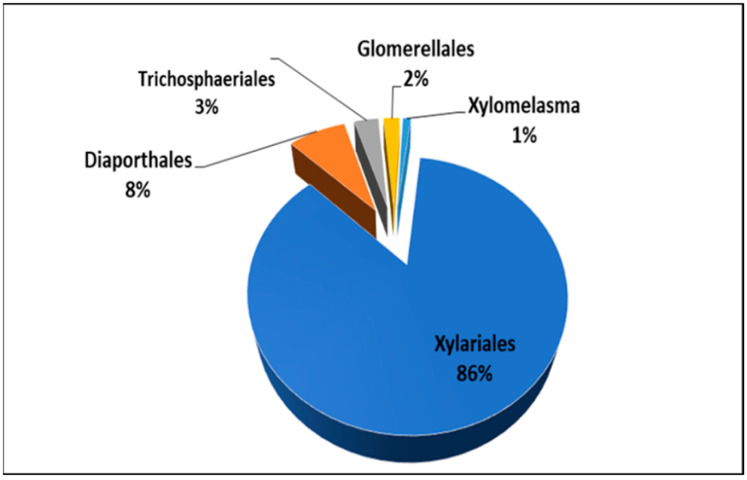
Relative abundance (%) of endophytic fungi in the class Sordariomycetes isolated from *Cornus florida* stems.

**Figure 3 plants-13-01250-f003:**
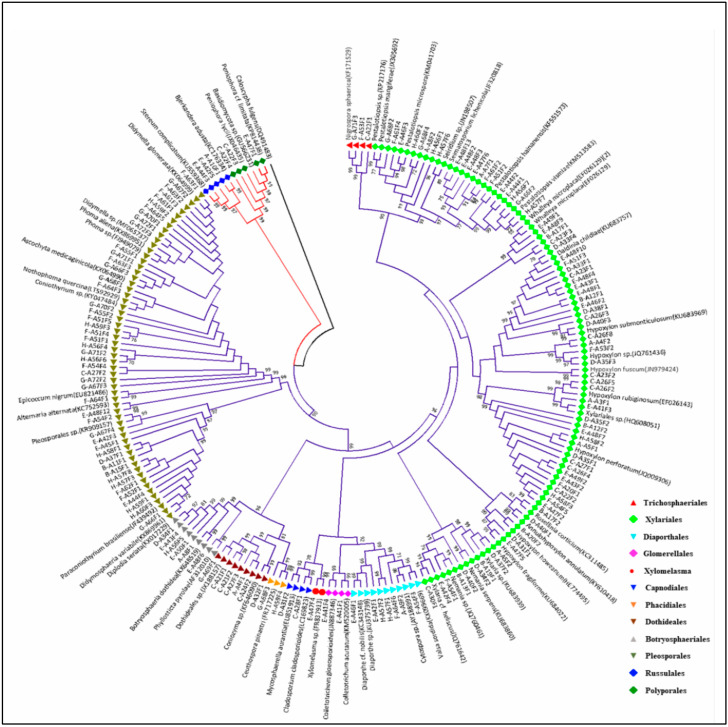
Phylogenetic relationships of endophytic fungi isolated from stems of healthy *Cornus florida* plants inferred using neighbor-joining method based on ITS sequences. The evolutionary distances were calculated using the p-distance method. Bootstrap values >70% as shown on branch obtained using 1000 bootstrap replicates. *Caloscypha fulgens* was used as outgroup taxa to root the tree.

**Figure 4 plants-13-01250-f004:**
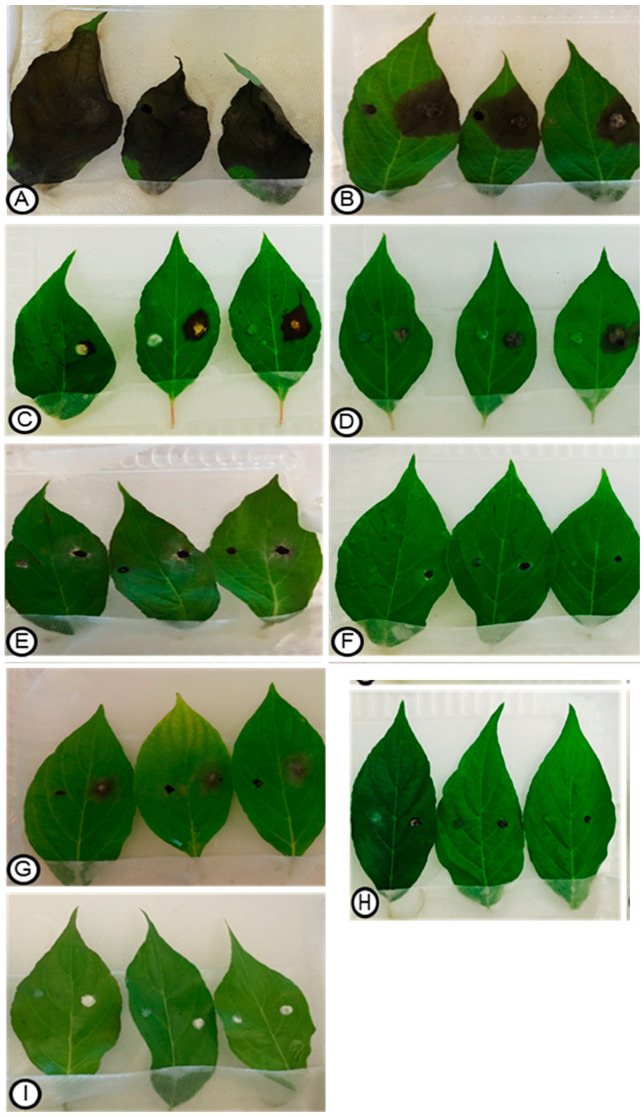
Necrotic lesions (**A**–**G**) produced on detached leaves of *Cornus florida* after inoculation where a 5.0-mm, mycelial disks of endophytic fungi were placed on half leaf (right of the midrib) as compared to mock inoculation with sterile PDA as control on the symmetrical half of the leaf (left of the midrib). Varying lesion sizes were produced by the following endophytes: (**A**) *Alternaria alternata* (E-A48F12); (**B**) *Nemania serpens* (DA40F2); (**C**) *Pestalotiopsis microspora* (A-A8F2); (**D**) *Botryosphaeria dothidea* (A-A8F1); (**E**) *Didymosphaeria variabile* (B-A11F1); (**F**) *Colletotrichum gloeosporioides* (E-A41F4); and (**G**) *Diplodia seriata* (E-A43F4), no necrotic lesions developed from the nonpathogenic isolates, *Cytospora* sp. (**H**)-[A-A9F1] and *Bjerkandera adusta* (**I**), [A-A10F1].

**Figure 5 plants-13-01250-f005:**
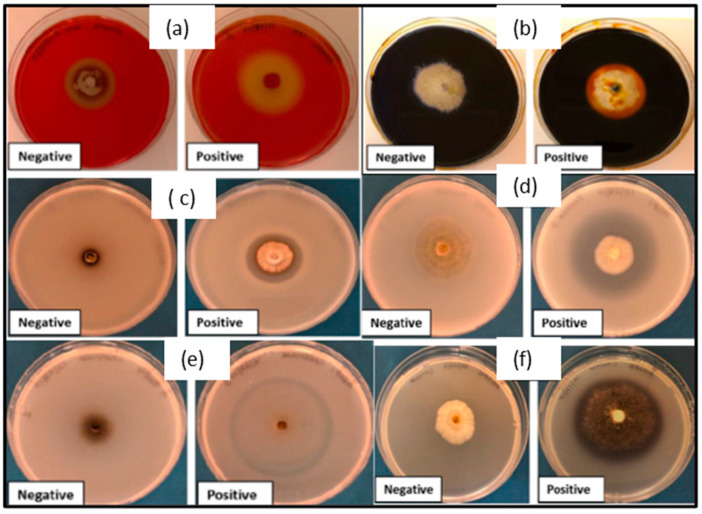
Screening for extracellular enzymes in fungal endophytes (derived from *Cornus florida*) showing a clear zone around respective colonies (positive reaction) compared to the control negative reaction (negative) indicating the production of (**a**) cellulase, (**b**) amylase, (**c**) pectinase, (**d**) protease, (**e**) chitinase, and (**f**) laccase (causing a blue color development).

**Table 1 plants-13-01250-t001:** Isolation frequency of endophytic fungi from Cornus florida stems from eight different locations and three counties in middle Tennessee, USA.

TN County	Warren					Davidson		Rutherford	Overall Total
Locations ^z^	A	B	C	D	E	F	G	H	8
Trees sampled	10	10	10	10	10	10	7	5	72
Tissue segments	150	150	150	150	150	150	105	75	1080
Total isolates	38	45	53	32	57	65	40	39	369
Number of isolates/tissues	0.25	0.3	0.35	0.21	0.38	0.43	0.38	0.52	

^z^ Locations A–H, in which location A–E are in McMinnville, F and G in Nashville, and H is a Murfreesboro location in Tennessee, USA.

**Table 2 plants-13-01250-t002:** Diversity indices of endophytic fungi isolated from *Cornus florida* growing at three counties in middle Tennessee.

Locations ^z^	A	B	C	D	E	F	G	H
Shannon Weiner index (H’)	2.06	1.89	2.434	2.155	2.673	2.61	2.216	2.5
Simpson’s diversity index (1-D)	0.87	0.84	0.904	0.913	0.912	0.926	0.874	0.931
Margalef’s richness (Dmg)	2.47	2.1	4.28	3.17	5.44	4.31	3.25	3.548
Plieou’s evenness index (J)	0.89	0.86	0.876	0.867	0.85	0.886	0.864	0.95
Menhinick’s index (Dmn)	1.62	1.34	2.197	2.12	3.046	2.36	2.06	2.242
Camargo’s index (1/Dmn)	0.62	0.75	0.455	0.47	0.328	0.424	0.485	0.446

^z^ Locations A–H, in which location A–E are in McMinnville, F and G in Nashville and H is a Murfreesboro location in Tennessee, USA.

**Table 3 plants-13-01250-t003:** Similarity values for endophytic fungal communities of *Cornus florida* growing at different locations.

Locations ^z^	B	C	D	E	F	G	H
A	0.174	0.235	0.214	0.233	0.293	0.08	0.143
B	-	0.242	0.276	0.2	0.176	0.083	0.148
C	-	-	0.22	0.235	0.22	0.171	0.167
D	-	-	-	0.22	0.205	0.074	0.133
E	-	-	-	-	0.25	0.053	0.245
F	-	-	-	-	-	0.304	0.299
G	-	-	-	-	-	-	0.3

^z^ Locations A–H, in which location A–E are in McMinnville, F and G are in Nashville and H is a Murfreesboro location in Tennessee, USA.

**Table 4 plants-13-01250-t004:** A list of fungal endophytes ^a^ that displayed pathogenic reaction in pathogenicity tests on detached *Cornus florida* leaves.

Botryosphaeria dothidea [A-A8F1]	Colletotrichum acutatum [E-A41F1]
Pestalotiopsis microspora [A-A8F2]	Didymosphaeria variabile [B-A11F1]
Rosellinia corticium [B-A17F2]	*Xylaria* sp. [B-A19F1]
Colletotrichum gloeosporioides [E-A41F4]	Diplodia seriata [E-A43F4]
Pestalotiopsis microspora [E-A44F1]	Hypoxylon perforatum [B-A20F1]
Nemania serpens [D-A40F2]	Alternaria alternata [E-A48F12]
Diplodia seriata [E-A50F1]	Phoma aliena [H-A56F4]
Daldinia childiae [F-A51F3]	Didymella glomerata [F-A61F1]

^a^ Endophytic fungi isolated from stem of symptom-free *Cornus florida* plants.

**Table 5 plants-13-01250-t005:** A list of endophytic fungi ^z^ that did not display disease reactions (nonpathogenic) during pathogenicity tests on *Cornus florida* leaves.

Nonpathogenic	Nonpathogenic	Nonpathogenic
*Hypoxylon* sp. [A-A4F2]	*Nigrospora sphaerica* [C-A22F1]	*Hypoxylon perforatum* [D-A35F2]
[E-A41F3]	*Hypoxylon perforatum* [C-A22F2]	*Hypoxylon* sp. [D-A35F]
*Hypoxylon perforatum* [A-A5F1]	*Peniophora lycii* [C-A22F3]	*Didymosphaeria variabile* [D-A37F1]
*Seimatosporium lichenicola* [A-A7F1]	*Hypoxylon fuscum* [C-A23F2]	*Xylaria* sp. [D-A37F2]
*Polyporales* sp. [A-A9F1]	*Pestalotiopsis* sp. [C-A24F1]	*Daldinia childiae* [D-A38F1]
*Cytospora* sp. [A-A9F2]	*Hypoxylon submonticulosum* [C-A26F3]	*Annulohypoxylon annulatum* [D-A40F1]
*Bjerkandera adusta* [A-A10F1]	*Hypoxylon perforatum* [C-A26F4]	*Hypoxylon submonticulosum* [D-A40F3]
*Daldinia childiae* [B-A12F1]	*Hypoxylon rubiginosum* [C-A26F5]	*Hypoxylon* sp. [D-A41F3]
*Hypoxylon rubiginosum* [B-A12F2]	*Ascochyta medicaginicola* [C-A27F2]	*Stereum complicatum* [E-A44F3]
*Nemania* sp. [B-A13F1]	*Cladosporium cladosporioides* [C-A28F1]	*Phyllosticta pyrolae* [E-A48F6]
*Didymosphaeria variabile* [B-A15F1]	*Xylaria* cf. *heliscus* [C-A30F1]	*Whalleya microplaca* [E-A48F9]
*Whalleya microplaca* [B-A17F1]	*Mycosphaerella aurantia* [D-A31F2]	*Didymella* sp. [F-A54F4]
*Hypoxylon howeanum* [B-A20F2]	*Coniozyma* sp. [D-A32F1]	*Coniothyrium* sp. [F-A55F1]
*Seimatosporium lichenicola* [C-A21F1]	*Daldinia childiae* [D-A33F1]	*Cytospora* sp. [H-A57F2]
*Dothideales* sp.[C-A21F2]	*Daldinia childiae* [D-A33F4]	*Ascochyta medicaginicola* [F-A61F3]
	*Diplodia seriata* [D-A34F1]	*Epicoccum nigrum* [F-A64F1]
	*Hypoxylon perforatum* [D-A35F1]	*Nothophoma quercina* [G-A70F2]
		*Didymella* sp. [G-A72F2]

^z^ Fungi isolated from stem of healthy appearing, symptom-free *Cornus florida* plant.

**Table 6 plants-13-01250-t006:** Locations of sites where samples were collected for fungal endophyte isolation from *Cornus florida* growing at three counties ^z^ in Mid-Tennessee.

Sites	Sample Collection in November 2014	
	Geographic Location	GPS Coordinates
A	McMinnville, TN	35°42′32.3″ N 85°44′42.0″ W
B	McMinnville, TN	35°42′28.5″ N 85°44′43.7″ W
C	McMinnville, TN	35°42′29.1″ N 85°44′45.5″ W
D	McMinnville, TN	35°42′33.6″ N 85°44′47.3″ W
E	McMinnville, TN	35°42′32.2″ N 85°44′37.3″ W
F	Nashville, TN	36°08′38.3″ N 86°40′15.3″ W
G	Nashville, TN	36°10′10.8″ N 86°47′12.1″ W
H	Murfreesboro, TN	35°52′49.3″ N 86°26′10.0″ W

^z^ Locations A–E in Warren County, F and G in Davidson County, and H in Rutheford County, Tennessee, USA.

## Data Availability

Data are contained within the article and Appendix A.

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
