# Peer review of "Endophytic Fungi Residing within Cornus florida L. in Mid-Tennessee: Phylogenetic Diversity, Enzymatic Properties, and Potential Role in Plant Health"

_plants, 2024, doi:10.3390/plants13091250_

Round 1
Reviewer 1 Report (Previous Reviewer 2)
Comments and Suggestions for Authors
The authors in their manuscript entitled “Endophytic fungi residing within Cornus florida L in mid-Tennessee: Phylogenetic diversity, enzymatic properties, and potential role on plant health”, present a study of the phylogenetic relationship and extent of diversity of endophytic isolates, along with pathogenicity testing and biochemical properties regarding hydrolytic enzymes produced.
The aim of research is clear, the methodological approach and implementation concerning the research presented are correct and the discussion of the results and subsequent conclusions support the main findings. The work is mainly descriptive, however, giving for the first time the background image of the endophytic populations of C. florida. Specific isolates are putatively of future interest regarding basic and applied research studies. In a future study the authors may wish to perform a Multi Locus Sequencing Analysis (Typing), using both existing ITS outcomes and results from other loci to be assessed, to reassure the identification/characterization of selected species under study.
Please enlarge headings and axis’ legends in Figure 1.
Author Response
Thank you so much for your review

Reviewer 2 Report (New Reviewer)
Comments and Suggestions for Authors
The study deals with an important topic – plant endophytes and their potential role in plant health.
The introduction is concise and the methodology is well-described. The results are generally presented well. Is Figure 2 necessary for understanding the results? At the same time, it would be useful to show the results of the enzyme production tests (the EI values for different endophyte taxa).
In the Discussion section the conclusion that some of the endophytes have potential benefits for plant health is based on two findings: some of the known pathogens had no effect on detached leaves and several fungi were shown to produce potentially anti-fungal enzymes. Is it possible that isolates of the potentially pathogenic fungi that did not infect the leaves were non-pathogenic because they have become associated with the host plant and different, more aggressive isolates of the same fungal species would have the pathogenic effect?
Is it possible to determine the species of Alternaria using the ITS sequence?
References 8, 14, 29 should be corrected (journal issue, pages should be stated, not an internet link).
In reference 23, in contrast, the internet link is lacking.
Other references should be checked as well
Ln 44 – perhaps, rephrase: “where these trees are planted”?
Author Response
Thank you so much for your review.

This manuscript is a resubmission of an earlier submission. The following is a list of the peer review reports and author responses from that submission.
Round 1
Reviewer 1 Report
Comments and Suggestions for Authors
-Material and Methods: it isn't clear what criteria were used to select the sample locations. The authors stated that they took samples from 8 locations, but it might be interesting for the reader to know about the different site conditions. It is well known (and noted by the authors as well) that the environment might have an influences on the diversity of endophytes, so which were the criteria for sample selection
- Please check additional literature e.g. Krabel, D., Morgenstern, K., & Herzog, S. (2013). Endophytes in changing environments - do we need new concepts in forest management? IForest - Biogeosciences and Forestry, 6(2), 109–112. https://doi.org/10.3832/ifor0932-006
The authors write: The objectives of this study were to (a) understand the diversity of fungal endophytes that reside in C. florida, evaluate the potential role of these fungal endophytes as plant pathogens and (b) biological control agents against selected diseases, and (c) explore the potential of these fungal endophytes to produce extracellular enzymes that may benefit plant health. To me it seems that objectives b and c (here: benefit for plant health) are not realy adressed. In my opinion these points should be further elaborated.
- Fig 1: a scale should be added to the pictures
Reviewer 2 Report
Comments and Suggestions for Authors
I really enjoyed the work and presentation. Although, a multi locus sequencing typing would (putatively) add more information regarding phylogenetic relationships, the authors may find the idea attracting for application in a future analysis regarding certain groups of endophytic fungi.